# An economic analysis of the health-related benefits associated with bicycle infrastructure investment in three Canadian cities

**David G. T. Whitehurst**[1,2]*, **Danielle N. DeVries**[3,4], **Daniel Fuller**[5,6], **Meghan Winters**[1,4]

**1** Faculty of Health Sciences, Simon Fraser University, Burnaby, BC, Canada, **2** Centre for Clinical Epidemiology and Evaluation, Vancouver Coastal Health Research Institute, Vancouver, BC, Canada, **3** Urban Studies Program, Simon Fraser University, Vancouver, BC, Canada, **4** Centre for Hip Health and Mobility, Vancouver Coastal Health Research Institute, Vancouver, BC, Canada, **5** School of Human Kinetics and Recreation, Memorial University of Newfoundland, Physical Education Building, St. John's, NL, Canada, **6** Department of Community Health and Humanities, Faculty of Medicine, Memorial University of Newfoundland, St. John's, NL, Canada

* david_whitehurst@sfu.ca

## Abstract

### Objectives

Decision-makers are increasingly requesting economic analyses on transportation-related interventions, but health is often excluded as a determinant of value. We assess the health-related economic impact of bicycle infrastructure investments in three Canadian cities (Victoria, Kelowna and Halifax), comparing a baseline reference year (2016) with the future infrastructure build-out (2020).

### Methods

The World Health Organization's Health Economic Assessment Tool (HEAT; version 4.2) was used to quantify the economic value of health benefits associated with increased bicycling, using a 10-year time horizon. Outputs comprise premature deaths prevented, carbon emissions avoided, and a benefit:cost ratio. For 2016–2020, we derived cost estimates for bicycle infrastructure investments (including verification from city partners) and modelled three scenarios for changes in bicycling mode share: 'no change', 'moderate change' (a 2% increase), and 'major change' (a 5% increase). Further sensitivity analyses (32 per city) examined how robust the moderate scenario findings were to variation in parameter inputs.

### Results

Planned bicycle infrastructure investments between 2016 and 2020 ranged from $28–69 million (CAD; in 2016 prices). The moderate scenario benefit:cost ratios were between 1.7:1 (Victoria) and 2.1:1 (Halifax), with the benefit estimate incorporating 9–18 premature deaths prevented and a reduction of 87–142 thousand tonnes of carbon over the 10-year time horizon. The major scenario benefit:cost ratios were between 3.9:1 (Victoria) and 4.9:1 (Halifax), with 19–43 premature deaths prevented and 209–349 thousand tonnes of carbon

**Data Availability Statement:** All relevant data are within the paper and its Supporting Information files.

**Funding:** DGTW, DF and MW received funding from the Canadian Institutes of Health Research (awards #365011 and #377333, https://cihr-irsc.gc.ca/e/193.html). MW was supported by a Michael Smith Foundation for Health Research Scholar Award (award 16502, https://www.msfhr.org/), and DF by a Canada Research Chair in Population Physical Activity (Government of Canada, https://www.chairs-chaires.gc.ca/home-accueil-eng.aspx). The funders had no role in study design, data collection and analysis, decision to publish, or preparation of the manuscript.

**Competing interests:** The authors have declared that no competing interests exist.

averted. Sensitivity analyses showed the ratio estimates to be sensitive to the time horizon, investment cost and value of a statistical life inputs.

## Conclusion

Within the assessment framework permitted by HEAT, the dollar value of health-related benefits exceeded the cost of planned infrastructure investments in bicycling in the three study cities. Depending on the decision problem, complementary analyses may be required to address broader questions relevant to decision makers in the public sector.

## Introduction

Bicycling in North America is not a widely used mode of transportation, despite the associated environmental and health benefits [1]. Health impact models of scenarios where transportation choices shift toward active modes indicate that, on balance, the physical activity benefits from walking and bicycling outweigh the risks associated with air pollution and traffic safety, across cities small and large, and across almost all age groups [2]. To encourage greater uptake in active transportation, cities are making major investments in bicycle infrastructure. Such investments can be considered population health interventions, i.e., real-world policy actions, often outside of the health sector, that shift the distribution of health risks [3].

Natural experiment studies of such real-world interventions employ diverse methodologies to answer questions most relevant given the context and the needs of decision-makers [4]. In particular, given the scarcity of public resources, decision-makers often request economic analyses–examinations of the costs and benefits of alternative courses of action–to rationalize investments. The scope of potential economic impacts (spanning congestion, business, accessibility, the environment, health, etc.) makes this a formidable undertaking. To date, transportation sector cost-benefit analysis frameworks have made limited effort to include health-related impacts beyond air pollution or injury risks [5]. Overlooking physical activity-related benefits of walking and bicycling underestimates the value of active transportation investments [6]. In a 2016 systematic review of economic analyses of active transportation interventions that incorporated physical activity (either implemented, proposed or hypothetical), only two of 29 (7%) studies for which the results were reported as ratios of benefits to costs indicated the respective costs outweighing the respective benefits [6]. Further generalizations in this field of research are hampered by the high level of heterogeneity in all aspects of evaluation (study design, quality and transparency, determination of costs and benefits, etc.). For example, in 25 of the 32 studies included in the systematic review by Brown and colleagues, the measurement of 'benefit' extended beyond the impact on physical activity, including a range of health (air pollution and injuries) and non-health (e.g., productivity, travel time, and congestion) factors [6].

An example of an attempt to incorporate impacts on health in an integrated model of transport scenarios was conducted by a research team in New Zealand, using participatory system dynamics modelling [7]. In collaboration with policy and community stakeholders, a context-specific causal model was generated to estimate the societal impacts of potential policy scenarios, acknowledging the complex feedback loops between infrastructure, health and environmental outcomes. Such rigorous undertakings require time, data and expertise beyond what city staff, transportation advocacy groups, and health and transportation practitioners can pursue. More user-friendly, stakeholder-oriented tools are becoming available, such as the World Health Organization's Health Economic Assessment Tool (HEAT) [8,9]. This tool has been

used to quantify the benefits of active travel (including assessments of current conditions or changes due to investments in active transportation) in European settings [10–12] and, less frequently, in Australia [13] and North America [14,15].

The study reported in this paper is part of the Impacts of Bicycle Infrastructure in Mid-Sized Cities (IBIMS) project. IBIMS is a natural experiment study, designed in collaboration with stakeholders, that is evaluating the impacts of a major investment to build an 'all ages and abilities' (AAA) bicycling network in Victoria, British Columbia (BC), Canada. Embracing the pragmatic paradigm of population health intervention research, IBIMS is evaluating changes in diverse outcomes (bicycling and bicycling safety, equity in spatial access, and economic impacts) in Victoria and two other Canadian cities [16]. Here, our objective is to estimate the health-related economic impacts of planned bicycle infrastructure investments in the three study cities. The paper also serves as a comprehensive and transparent account of using HEAT, detailing data sources and assumptions for inputs that may be particularly relevant for researchers and practitioners in the North American context.

## Methods

The IBIMS project received approval from the Simon Fraser University Office of Research Ethics (study number 2016s0401). Survey participants in the IBIMS project provided written consent prior to participation [16].

### Health economic assessment tool

The World Health Organization's HEAT (version 4.2, released May 2019) is used to quantify the economic value of benefits to health that result from bicycling or walking [8,9]. HEAT can be used to assess point-in-time scenarios (e.g., existing states of affairs) using data from a single timepoint, or the impact of an investment/intervention using before-and-after data. Data on population-level physical activity, air pollution exposure, and transportation-related injuries are used to generate estimates of premature deaths prevented, carbon emissions avoided, and health-related economic impacts. In this context, 'health-related economic impacts' can refer to the monetary value of the premature deaths prevented and carbon emissions avoided (i.e., the 'benefits'), or the benefit:cost ratio (where 'cost' is the infrastructure investment).

The number of input parameters required to perform an analysis using HEAT version 4.2 is dependent on the study objective and/or data availability. The tool starts with inputs for 'Your Assessment', which include general information about the transportation mode being assessed (walking, bicycling or both), geographic scale (country level, city level or sub-city level), type of assessment (single case or two-case), the time scale over which the impacts of investment should be calculated, and what impacts to assess (physical activity, air pollution, crash risk, and/or carbon emissions). This information determines the 'use case' and the subsequent methodology and assumptions that the tool applies. The next stage requires the user to work through several sections of data inputs, including per-person, per-day estimates for the relevant travel modes. The tool provides default entries for several parameters, both categorical (e.g., the population to be assessed is of working age, defined as 20–64 years) and numerical (e.g., average bicycling speed is 14.0 kilometres (km) per hour). All default entries can be adjusted to meet the user's context, which is an important feature given that HEAT was originally designed for a European audience.

### Study cities & bicycle infrastructure investments

Comprehensive details of the IBIMS project are described elsewhere [16]. The study cities are Victoria (BC), Kelowna (BC) and Halifax (Nova Scotia). Based on guidance by study partners,

our definition of the 'study city' for Victoria includes the neighboring municipalities of Saanich, Oak Bay and Esquimalt. For Halifax, the study city includes the downtown peninsula, mainland metropolitan core and Dartmouth. The study city for Kelowna includes the entire municipality. City-specific baseline data on population size, mode share, and bicycle infrastructure are provided in Table 1.

The scale of infrastructure investment between 2016 and 2020 varies across study cities. In Victoria, investments relate to the AAA bicycle network, which aims to build 32 km of new safe bicycling routes (on top of the existing 194 km of bicycling infrastructure in place) to connect the downtown core to residential neighbourhoods, employment centres, schools, parks and recreation centres. Victoria is also implementing new crosswalks, shared paths and 'complete streets' (streets designed explicitly for all transportation modes, and people of all ages and abilities) [21]. Kelowna is implementing a shared path for bicycling along a discontinued rail corridor and adding ~20 km of active travel corridors in the downtown core [22]. Halifax is adding a bikeway to the primary bridge between the downtown peninsula and Dartmouth, and other active travel projects like sidewalk renewal [23].

## Analysis & data inputs

The city-level HEAT analyses reported in this paper are two-case assessments, comparing a baseline reference year (2016) with the future infrastructure build-out (2020), quantifying the economic value of health benefits of bicycling only. Analyses are conducted from the perspective of the respective study cities. We modelled the health impacts of bicycling over a 10-year time horizon–the HEAT default period over which the discounted mean annual benefit is calculated–considering all potential impacts (physical activity, air pollution, crash risk and carbon emissions) at the city level. Each analysis comprises a combination of numerical entries and categorical selections. Details of all parameters and inputs are described in the following subsections and/or reported across Table 2 and S1 Table. To maximize transparency, Table 2 and SM1 use the same category headings used in the tool ('Your Assessment' (SM1 only), 'Data Input', 'Data Adjustment', 'Monetization Parameters' and 'Parameter Review') and parameter labels that reflect the wording used in the respective subsections of HEAT v4.2. [N.B. the authors are aware that changes have been made to HEAT 4.2 without any corresponding

**Table 1. Baseline (2016) population, mode share and bicycle infrastructure characteristics of the study cities.**

|  | Victoria | Kelowna | Halifax |
|---|---|---|---|
| Population (all ages)[a] [17] | 235,689 | 128,669 | 204,927 |
| Study area (km$^2$) [17] | 141 | 214 | 122 |
| Population density (people/km$^2$) [18–20] | 1,673 | 602 | 1,675 |
| Bicycling mode share[b] [17] | 8.7% | 3.7% | 1.7% |
| Driving mode share[b] [17] | 62.7% | 83.8% | 65.8% |
| Bicycling infrastructure (km)[c] [17] |  |  |  |
| Cycle track | 3 | 5 | 1 |
| On-street bicycle lane | 92 | 157 | 36 |
| Off-street path (bicycle only or multi-use) | 73 | 81 | 45 |
| Local street bikeway | 27 | 0 | 1 |
| Total | 194 | 243 | 82 |

[a] These data are at the census dissemination area, aggregated to the study city areas.

[b] Reported main mode of commuting for the employed labour force aged 15 years and over.

[c] As provided by IBIMS city partners (January 2017), with all estimates reported to the nearest kilometre.

**Table 2. Parameters and input data, including source, for the primary two-case (2016, 2020) assessments of bicycling infrastructure investment over a 10-year time horizon[a].**

| Parameter | Input | | |
|---|---|---|---|
| | Victoria | Kelowna | Halifax |
| Data Input–Active Modes Data | | | |
| Baseline (2016) bicycling (min./person/day)[b] [24] | 4.1 | 1.4 | 0.9 |
| Bicycling in 2020: 'no change' (0%) scenario (min./person/day)[c] | 4.1 | 1.4 | 0.9 |
| Bicycling in 2020: 'moderate change' (2%) scenario (min./person/day)[d] | 5.3 | 2.4 | 1.6 |
| Bicycling in 2020: 'major change' (5%) scenario (min./person/day)[d] | 7.0 | 3.8 | 2.6 |
| Population 2016 (number)[b] [18–20] | 154,848 | 84,021 | 142,014 |
| Population 2020 (number)[d] [18–20] | 161,042 | 89,734 | 145,991 |
| Data Input–Motorized Modes Data | | | |
| Baseline (2016) driving (min./person/day)[b] [24] | 38.4 | 51.0 | 41.7 |
| Driving in 2020: 'no change' (0%) scenario (min./person/day)[c] | 38.4 | 51.0 | 41.7 |
| Driving in 2020: 'moderate change' (2%) scenario (min./person/day)[d] | 37.3 | 49.8 | 40.6 |
| Driving in 2020: 'major change' (5%) scenario (min./person/day)[d] | 35.7 | 48.1 | 39.1 |
| Baseline (2016) public transport (min./person/day)[b] [24] | 7.3 | 2.5 | 6.4 |
| Public transport in 2020 (for all scenarios) (min./person/day)[c] | 7.3 | 2.5 | 6.4 |
| Data Adjustment–General Adjustments | | | |
| Bicycling data excluded due to other interventions (%)[c] | 0 | 0 | 0 |
| Temporal and spatial adjustment (%)[c] | 0 | 0 | 0 |
| Take-up time for new bicycling (years)[d] | 4 | 4 | 4 |
| Data Adjustment–Contrast Characteristics (%) | | | |
| New trips[c] | 0 | 0 | 0 |
| Bicycling for transport (as opposed to recreation)[d] [24] | 98 | 98 | 99 |
| Data Adjustment–Other Adjustments | | | |
| Bicycling in traffic (%)[d] [17] | 49 | 67 | 44 |
| Traffic conditions[d,e] | 'some' | 'some' | 'some' |
| Change in crash risk (%)[c] | 0 | 0 | 0 |
| Substitution of physical activity (%)[c] | 0 | 0 | 0 |
| Monetization Parameters | | | |
| Investment cost (million $)[b,f] [25–28] | 68.7 | 27.9 | 28.7 |
| Discount year[d] | 2016 | 2016 | 2016 |
| Parameter Review–Calculation Parameters | | | |
| Carbon value 2016 ($/tonne)[d,f] [29] | 43.0 | 43.0 | 43.0 |
| Carbon value 2025 ($/tonne)[d,f] [29] | 52.5 | 52.5 | 52.5 |
| Discount rate (%)[d] [30] | 1.5 | 1.5 | 1.5 |
| Average bicycling speed (km/h)[c] | 14.0 | 14.0 | 14.0 |
| Average car speed (km/h)[c] | 42.0 | 42.0 | 42.0 |
| Average public transport speed (km/h)[c] | 22.7 | 22.7 | 22.7 |
| Value of a statistical life (million $)[d,f,g] [31] | 6.5 | 6.5 | 6.5 |
| PM2.5 concentration (µg/m$^3$)[d] [32,33] | 3.9 | 5.9 | 4.9 |
| All-cause mortality rate 2016 (deaths/100,000 people)[d] [34,35] | 228.7 | 228.7 | 228.7 |
| All-cause mortality rate 2020 (deaths/100,000 people)[d] [34,35] | 228.7 | 228.7 | 228.7 |
| Bicycling fatality rate (fatalities/hundred million km)[d,h] [36] | 2.23 | 2.23 | 2.23 |

[a] All inputs are reported as they were entered into the tool. With the exception of the discount rate, HEAT requires all percentages to be entered as integers.

[b] User is required to enter a response (i.e., there is no default entry in HEAT).

[c] Parameter has a default entry in HEAT and the default entry was used. Some default entries are populated based on responses to earlier questions. For example, the min./person/day parameters for 2020 were populated with default entries that were the same numbers as entered for 2016.

[d] Parameter has a default entry in HEAT and the default entry was changed.

[e] Five input options: European average in urban areas; free flow; some congestion; heavy congestion; European average in rural areas.

[f] Monetary inputs are reported in Canadian dollars. HEAT requires inputs (and provides outputs) in euros; we used an exchange rate of $1:€0.65 (January 2016; https://www.xe.com/).

[g] A 2009 estimate for the Canadian population aged under 65 years.

[h] Estimate confirmed through personal communications with the author of peer-reviewed research on the BC-specific fatality rate.

change to the version number. Details reported in this paper relate to analyses that were conducted in version 4.2 in May 2020].

**Data input–modes data (active and motorized).**   Bicycling mode share, i.e., the percentage of total trips made by bicycle, is a common metric used by transportation planners. Three scenarios of mode share change between 2016 and 2020 were considered: no change (i.e., 0% increase in bicycling mode share), moderate change (a two percentage-point increase), and major change (a five percentage-point increase). The inputs for 'modes data' in HEAT are in a minutes/person/day metric. To obtain these estimates we used data from our 2016 baseline population survey [24], using responses from participants aged 18–64 years (the closest match to the 20–64 age range that is used in HEAT). The survey asked about the purpose, transportation mode and length of each trip in a one-day trip diary. For each mode, we converted these data into the compatible metric by multiplying the total number of trips by the average length of trip (in minutes) and dividing by the number of respondents. To calculate the minutes/person/day estimates for the mode shift scenarios, the 'total number of trips' was amended to reflect the scenario-specific mode share percentages. For illustration, the 2% increase in bicycling mode share in Victoria (from 8.7% in 2016 (see Table 1) to 10.7% in 2020) corresponded to an increase of 1.2 minutes of bicycling per person, per day (see Table 2). We assumed that all new bicycling trips were diverted from driving trips, meaning that a 2% increase in bicycling mode share (e.g., from 8% to 10%) corresponded to a 2% decrease in car mode share (e.g., 62% to 60%). Baseline estimates for public transport were held constant across the study period.

The three scenarios (0%, 2% and 5%) were chosen as realistic mode share changes at the population-level. Previous studies have found effects ranging from 0% to 5% regarding the increase in the amount of people bicycling for transportation after an intervention and modelled anywhere from 1% to 50% of car trips being replaced by bicycle trips [2,37]. Our study is in mid-sized Canadian cities, designed predominantly for car use, and where baseline bicycling mode share is low [38].

Population estimates in each study city were identified using the 2016 Census dissemination areas that correspond to the municipal study areas (spatial methods described elsewhere [17]). Each population was adjusted to a working-age population (15–64 years, the closest match to the 20–64 age range that is used in HEAT), using the percentage of the total population in this category from the census profiles [18–20]. We estimated the 2020 population using the annual percentage increase over the previous census period (2011–2016) [18–20].

**Data adjustment–general adjustments.**   For this application of HEAT, it was assumed that there were no other factors (i.e., independent to the interventions being assessed) that would lead to a change in bicycling. We also assumed no temporal or spatial adjustments were needed because our travel data was collected in fall, which can be considered typical travel behaviour [39]. Since the two cases (timepoints) in our study are four years apart (2016 and 2020), and new bicycle infrastructure is continually built out during this time, we assumed that the increase in bicycling mode share took the full four years to reach the level projected for 2020. Using a 10-year time horizon (the default setting in HEAT) means that the scenario-specific increase in bicycling mode share was realized through a constant linear increase over the first four years, then the mode share plateaued for the remaining six years.

**Data adjustment–contrast characteristics.**   The model assumed no new trips over the study period. We calculated the percentage of all bicycling trips that were for transportation based on the trip purpose question in our 2016 population survey [24].

**Data adjustment–other adjustments.**   We calculated the proportion of bicycling that is exposed to traffic (i.e., those at a higher crash risk and higher exposure to air pollution) using the relative length of types of bicycle infrastructure observed in each city (spatial methods described elsewhere [17]). This was calculated using the kilometres of infrastructure exposed

to major roads (bicycle tracks and painted lanes) relative to the total length of infrastructure in the city. For the traffic conditions parameter, we selected 'some congestion' (one of the five input options) in all three cities. We assumed there was no change in crash risk, and that any increase in bicycling between 2016 and 2020 was not the result of a decrease in other forms of physical activity (i.e., there is no substitution effect).

**Monetization parameters.** To populate the investment costs parameter, we extracted data from publicly available financial plans and active transportation plans for each of the three study cities [25–28]. Specifically, we sought information about the costs of (i) capital projects for new active transportation infrastructure inclusive of bicycling, sidewalk, and active travel/ complete street corridors, and (ii) maintenance of existing infrastructure. All modes of active transportation were included, since some cities had budget lines attached to active corridors rather than bicycle-specific infrastructure. We used 2017 financial plans and captured 2017 capital costs, as well as 2017–2020 operating costs associated with the capital projects. Once initial estimates were derived, we contacted our city partners (planners in the City of Kelowna and Halifax Regional Municipality, and all four municipalities in the Victoria study area) to confirm that our estimates were reasonable approximations of the intended costs. Our conversations with city partners highlighted that costs in budgets were estimates and averages, with the inevitability of variation from year to year. Total cost estimates were discounted to the baseline year (2016), using a discount rate of 1.5% [30].

**Parameter review–calculation parameters.** HEAT provides default entries for all calculation parameters based on selections made while entering data inputs. Except for average speed by mode, we changed all default entries. Canada-specific inputs for the social cost of carbon, discount rate, and value of a statistical life were identified from openly available sources [29–31]. All provinces track air quality metrics, so the PM2.5 concentrations for each city were extracted for 2016 from provincial data repositories [32,33]. The all-cause mortality rate was calculated for the working-age population (15–64 years, the closest match to the 20–64 age range that is used in HEAT), using administrative data and age-standardization rates from Statistics Canada [34,35]; the same rate was used for 2016 and 2020. Finally, we confirmed the Canada-wide bicycling fatality rate with the author of peer-reviewed research on the BC-specific fatality rate [36].

For each scenario, we report (i) the number of premature deaths prevented for each of the impact pathways (physical activity, air pollution, and crash risk), (ii) the amount of carbon saved and (iii) the benefit:cost ratio. For all benefit:cost ratios reported in this paper, the consequent term (i.e., the cost term) is rescaled at one. We conducted univariate sensitivity analysis to explore the robustness of results to variations in 18 parameter inputs (32 analyses per city, in total), details of which are provided in S2 Table. All sensitivity analyses were performed using the bicycling and driving mode share estimates for the moderate change (2% increase) scenario. The impact of the changes in parameter inputs on the city-specific benefit:cost ratios are presented using tornado diagrams, with the changes in the antecedent term (i.e., the benefit term) displayed relative to the results of the respective primary analyses. A 'change' in the benefit term was defined as an absolute difference equal to or greater than 0.05 when comparing the results of the sensitivity analysis with the primary analysis.

## Results

Table 3 reports the total number of premature deaths prevented, premature deaths prevented by impact pathway, carbon saved, and benefit:cost ratios, under each change in bicycling mode share scenario (over the 10-year time horizon). Annual estimates of premature deaths prevented, by impact pathway, are illustrated in Fig 1. We see that the benefit:cost ratios are

**Table 3. Number of premature deaths prevented by impact pathway (physical activity, air pollution, crash risk), carbon saved, and benefit:cost ratios for each change in bicycling scenario, by city, over the 10-year time horizon[a].**

| Output | City and Scenario (% increase in bicycling mode share) | | | | | | | | |
|---|---|---|---|---|---|---|---|---|---|
| | Victoria | | | Kelowna | | | Halifax | | |
| | No change (0%) | Moderate change (2%) | Major change (5%) | No change (0%) | Moderate change (2%) | Major change (5%) | No change (0%) | Moderate change (2%) | Major change (5%) |
| Premature deaths prevented (total) | 1.9 | 18.4 | 42.5 | 0.7 | 8.6 | 19.1 | 0.3 | 8.6 | 21.1 |
| Number due to physical activity | 2.0 | 19.0 | 44.0 | 0.7 | 9.0 | 20.0 | 0.3 | 9.0 | 22.0 |
| Number due to air pollution exposure | -0.04 | -0.3 | -0.8 | -0.02 | -0.3 | -0.6 | -0.01 | -0.2 | -0.5 |
| Number due to crashes | -0.04 | -0.3 | -0.7 | -0.01 | -0.1 | -0.3 | -0.01 | -0.2 | -0.4 |
| Carbon saved (tonnes, thousands) | 0.0 | 142.2 | 349.0 | 0.0 | 86.5 | 209.1 | 0.0 | 129.1 | 305.2 |
| Benefit:cost ratio[b] | 0.2:1 | 1.7:1 | 3.9:1 | 0.1:1 | 1.9:1 | 4.3:1 | 0.1:1 | 2.1:1 | 4.9:1 |
| Economic benefit (million $)[c] | 12.8 | 116.6 | 264.6 | 4.0 | 52.5 | 120.5 | 1.8 | 59.1 | 140.3 |
| Cost (million $)[c] | 68.7 | 68.7 | 68.7 | 27.9 | 27.9 | 27.9 | 28.7 | 28.7 | 28.7 |

[a] Estimates are reported to either one decimal place or one significant figure.

[b] The benefit:cost ratios reported in this table were calculated to one decimal place for the benefit term, using the economic benefit estimate generated by HEAT and the city-specific cost estimates.

[c] Monetary inputs are reported in Canadian dollars. HEAT requires inputs (and provides outputs) in euros; we used an exchange rate of $1: €0.65 (January 2016; https://www.xe.com/).

similar, within scenarios, across the three cities. As would be expected, investment costs outweigh benefits in the 'no change' scenario (benefit:cost ratios of 0.2:1 in Victoria, and 0.1:1 in Kelowna and Halifax). For the moderate scenario (a 2% increase in bicycling mode share), the estimated benefit:cost ratios are close to 2:1 (1.7:1 in Victoria, 1.9:1 in Kelowna, and 2.1:1 in Halifax), capturing approximately 18 premature deaths prevented in Victoria, nine in Kelowna and nine in Halifax. Under the major scenario (a 5% increase in bicycling mode share), the benefit:cost ratios are 3.9:1 in Victoria, 4.3:1 in Kelowna, and 4.9:1 in Halifax, with 43, 19 and 21 premature deaths prevented in Victoria, Kelowna and Halifax, respectively. Physical activity due to bicycling is the main component of the estimated health-related impacts; the impacts of air pollution and traffic crashes have only minor influence, with estimates lower than one premature death over 10 years across all analyses. Regarding carbon emissions, the moderate scenario was associated with estimated savings of between 87 (Kelowna) and 142 (Victoria) thousand tonnes of carbon over the 10-year period, while the major scenario resulted in estimates ranging from 209 (Kelowna) to 349 (Victoria) thousand tonnes.

Findings from the sensitivity analyses showed changes in the benefit:cost ratio in 17 (53%), 17 (53%) and 20 (63%) of the 32 analyses performed for Victoria, Kelowna and Halifax, respectively. Fig 2 illustrates the changes in the benefit terms across the three study cities; results for all sensitivity analyses are provided in S3 Table. Consistent observations were that the benefit:cost ratio was sensitive to the modelled changes for the time horizon, investment cost and value of a statistical life parameters. The benefit term fell below 1.0 in all three cities when the time horizon of the study was reduced to five years. The benefit term also fell below 1.0 when doubling the investment costs in Victoria and Kelowna, and when reducing the estimate for the value of a statistical life by 50% in Victoria.

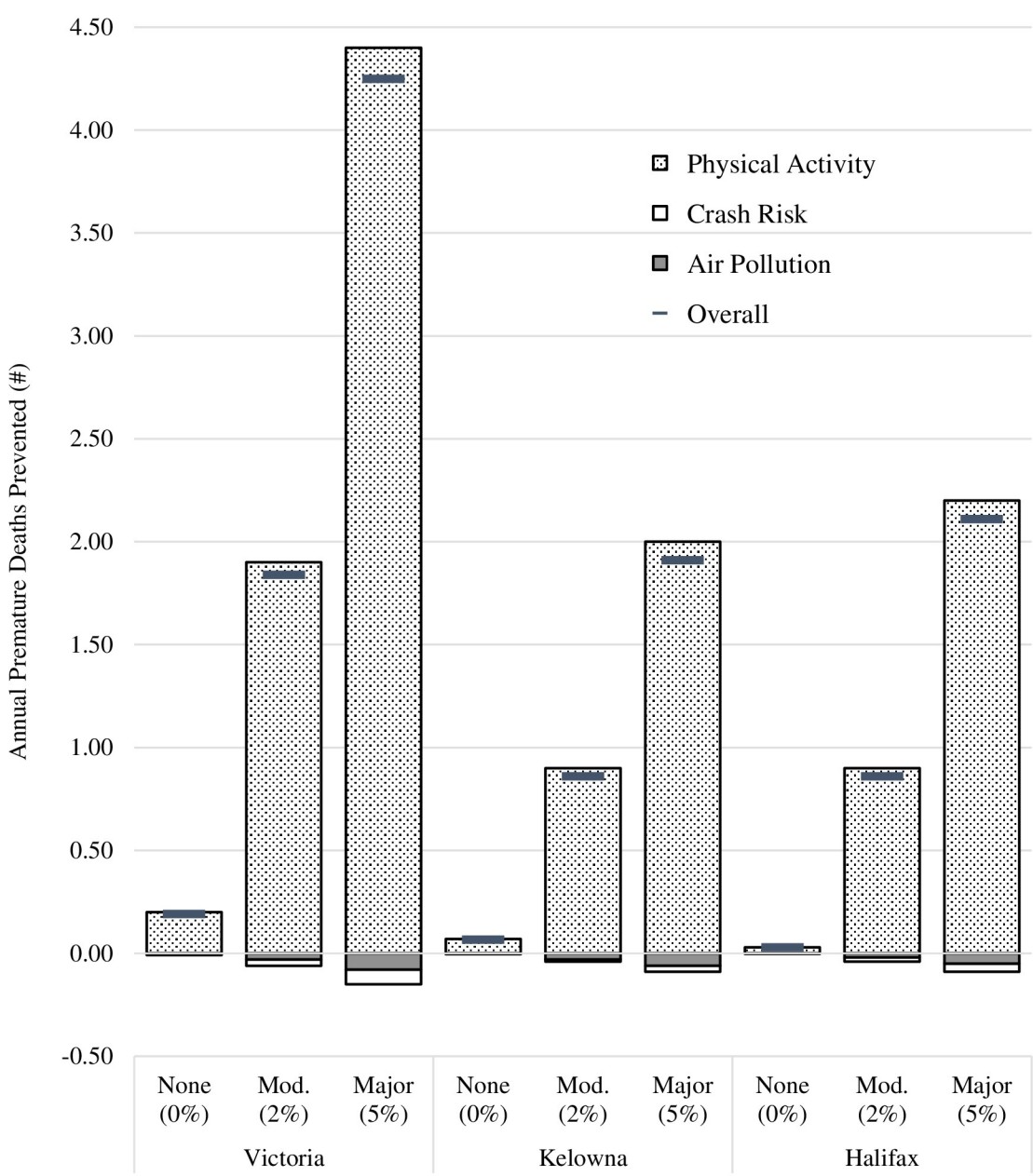

Change in Bicycling Mode Share Scenarios, by City

**Fig 1. Annual estimates of the number of premature deaths prevented under each bicycling mode share scenario, by pathway (physical activity, air pollution and crash risk) and city, for the 10-year time horizon of the study.**

## Discussion

In this study, we used the World Health Organization's HEAT (version 4.2) to quantify the economic value of health benefits resulting from bicycle infrastructure investments in three Canadian mid-sized cities (Victoria, Kelowna and Halifax). With outputs expressed as a benefit:cost ratio, our findings were similar across the cities. Estimates were between 1.7:1 (Victoria) and 2.1:1 (Halifax) when modelling a moderate change (a 2% increase) in bicycling mode

*Panel A (Victoria)*

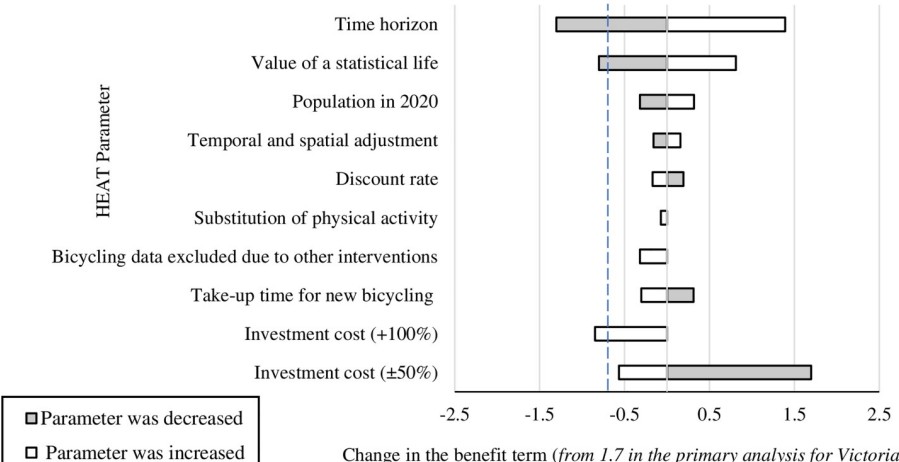

Change in the benefit term (*from 1.7 in the primary analysis for Victoria*)

*Panel B (Kelowna)*

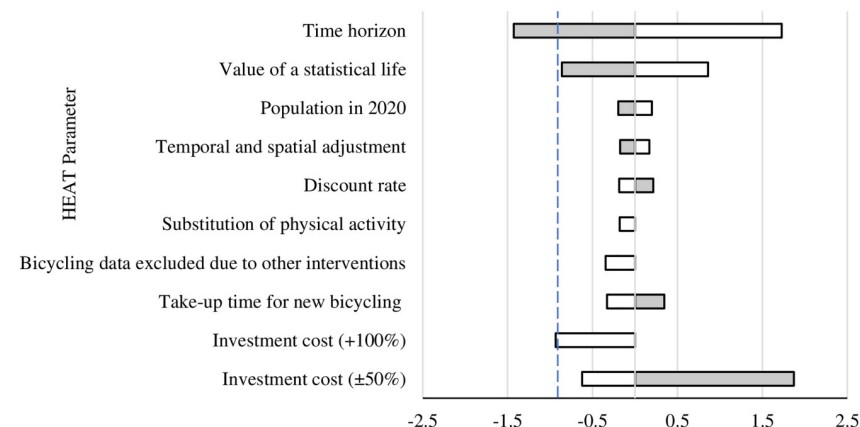

Change in the benefit term (*from 1.9 in the primary analysis for Kelowna*)

*Panel C (Halifax)*

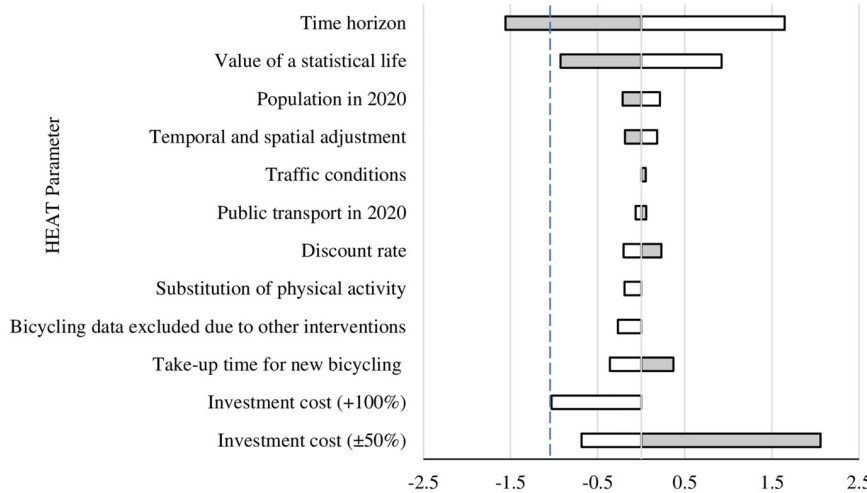

Change in the benefit term (*from 2.1 in the primary analysis for Halifix*)

**Fig 2. Findings from the univariate sensitivity analyses for the moderate change (2% increase) scenario in Victoria (Panel A), Kelowna (Panel B) and Halifax (Panel C), where variation in inputs led to a change in the respective benefit:cost ratio. Details of the changes made to the input parameters are provided in SM2.** [a] [a] A 'change' in a ratio estimate is defined as an absolute difference for the benefit term equal to or greater than 0.05. The results of all sensitivity analyses (to one decimal place) are reported in SM3. In each panel, the dashed line indicates the point where the benefit:cost ratio is 1:1.

share over a 10-year time horizon, and between 3.9:1 (Victoria) and 4.9:1 (Halifax) when modelling for major changes (a 5% increase), which means that the dollar value of health-related benefits exceeded the cost of planned infrastructure investments in all primary assessments. Given the differential timings of costs and benefits for such infrastructure investments (i.e., upfront costs and a long-term view required to realize the benefits), the benefit:cost ratio is a useful metric for assisting municipal staff and decision-makers to assess the consequences of investment decisions.

Within the HEAT framework, health benefits are measured in terms of premature deaths prevented (from physical activity, air pollution and crash risk) and the amount of carbon saved, with outputs provided in both natural units and dollar values. In our study cities, the major determinant of the economic benefit resulted from the prevention of premature deaths due to physical activity; in comparison, increases in premature deaths due to more people being exposed to crash risks and air pollution were small (see Table 3 and Fig 1). In cities with higher levels of air pollution or greater road safety challenges, the detrimental impacts may counter the benefits from physical activity to a greater degree. However, apart from one study from Belgium and Flanders, a review of diverse health impact modelling studies across Europe, India, the United States and Australasia reported that physical activity contributed to at least 50% of all estimated health impacts [2]. Some of these health impact modelling studies have used more nuanced tools, such as the Integrated Transport and Health Impact Modelling Tool (ITHIM) [40] or the Impacts of Cycling Tool [41], which can generate age- and sex-specific impacts, and also enable modelling of morbidity outcomes using disability-adjusted life years. While these outputs may be desirable for some decision-makers, such tools require detailed data (e.g., disaggregated travel data and age-specific risk estimates). None of these tools has been applied in Canada to date, and they require inputs not widely available in the Canadian context.

Against a backdrop of scarce resources, it should be expected that decision makers (whether local, regional or national) will demand analyses that examine the economic consequences of potential interventions. Similarly, with physical inactivity being a leading cause for the growing prevalence of non-communicable diseases, it is important to examine the economic implications of interventions that aim to increase physical activity [42]. The utility of a stakeholder-oriented tool, such as HEAT, can be seen in the range of outputs provided. In addition to the benefit:cost ratio and estimates of premature deaths prevented, the estimates of carbon saved (based on the assumption that trips are made by bicycle instead of car) may be valuable in the context of cities' sustainability targets. The tool has rich documentation and guidance, but end users must make informed decisions about relevant inputs. Our detailed description and disaggregated presentation of inputs was provided, in part, to give readers an illustration of the data requirements and the flexibility of the tool regarding context-specific inputs (versus default entries). Sensitivity analysis also provides a means of exploring variation in HEAT outputs that result from changes in parameter inputs. For example, the findings in this study were robust to variation in inputs. The cost of infrastructure investments only exceeded (or came close to exceeding) the dollar value of health-related benefits in scenarios where investment costs were doubled, the value of a statistical life was halved, or a significantly shorter time

horizon was considered. The identification of plausible ranges of variation for HEAT inputs is an important step in the development of an analysis plan.

It is common for terms such as 'economic analysis', 'economic evaluation' and 'cost-benefit analysis' to be used as generalizations for analyses that look to examine the costs and benefits of a particular course of action. Clarity is important when considering a specific technique for the purposes of supporting transparent and rigorous decision-making. As discussed in a systematic review of economic analyses of active transport interventions [6], Brown and colleagues highlight that the prevalence of different types of economic evaluation varies across sectors, with cost-benefit analysis being the traditional form of analysis within the transport sector, and cost-utility analysis dominating in the health sector [30,43]. HEAT analyses more closely align with cost-benefit analysis, where the benefits of an intervention are expressed in monetary terms and compared with the costs of the intervention. However, analyses using HEAT should not be regarded as full, comprehensive economic evaluations (nor should analysts claim so); the tool was developed as a means of estimating the value of reduced mortality, which can complement other techniques for economic assessment. For this reason, it is important to understand similarities and differences between HEAT and other forms of economic assessment (although a full exposition is beyond the scope this paper). One key issue is the comparators under consideration. By definition, economic evaluation is an incremental analysis, meaning that it is necessary to examine the *additional* costs and *additional* benefits of one course of action relative to alternatives (including the *status quo*, which is akin to 'usual care' in a health care context). In a single-case HEAT assessment, there is an implicit comparison of 'no walking or bicycling', which is unlikely to be a meaningful comparator when viewed within a cost-benefit or cost-utility framework. In situations where two-case HEAT assessments have been conducted, Brown and colleagues identify significant shortcomings in the reporting of comparison scenarios [6]. Regardless of the means of economic assessment, accurate and transparent descriptions of comparators are important features of any analysis that has the objective to facilitate evidence-based decision-making.

The strengths of our work lie in: (i) the contribution to population health intervention research, where economic analyses are infrequently performed but widely sought after; (ii) the conduct of research in cities that would rarely have the data or capacity to undertake such analysis; (iii) the transparent and disaggregated detailing of our inputs, supporting the use of HEAT in other settings; (iv) the collaboration with stakeholders to best understand infrastructure costs to support bicycling; and (v) the comprehensive sensitivity analyses.

As with all studies, there are limitations and necessary cautions. First, regarding the assessment tool, it is important to note that analysis with HEAT comprises population-level data inputs and outputs, habitual behaviours (rather than infrequent, recreational habits), mortality effects only, and assumes behavioural changes are within a working-age adult population of 20–64 years. Second, given the number of estimates that are used in the tool, it is inevitable that there is a degree of uncertainty. This is reflected in the 2017 HEAT User Guide recommendations, which state that, "*uncertainties around an assessment be made explicit and that the calculations be carried out with high and low estimates of the main variables to improve the understanding of the possible range of the final results.*" [8]. The 32 city-specific univariate sensitivity analyses described in SM2 were devised by the study team; this is a subjective exercise and different analysts are likely to have devised different scenarios. Finally, considerations such as equity (i.e., who are the beneficiaries of infrastructure investment) and the impact of active transportation on quality of life are not captured in the tool. While this is not a limitation of the current study (the study is an application of HEAT), it is important to recognize that economic assessments are likely to require additional analyses (requiring different methodologies) to complement HEAT, depending on the scope of the decision problem or policy

question. For example, cities and stakeholders may be interested in economic impacts of new bicycling infrastructure that are beyond health (e.g., business-related or tourism impacts) or the implications associated with technology advances such as hybrid and autonomous vehicles.

## Conclusion

Within the assessment framework permitted by HEAT, the dollar value of health-related benefits exceeded the cost of planned infrastructure investments in bicycling in the three study cities (all mid-sized Canadian cities). HEAT is a useful method to monetize health-related benefits associated with active transportation infrastructure. This study provides an important contribution to advance this interdisciplinary, intersectoral space by considering the utility of a practitioner-oriented tool in an applied setting. Depending on the decision problem, it is likely that complementary analyses would be required to address broader questions relevant to decision makers in the public sector.

## Supporting information

**S1 Table. Inputs for the parameters not reported in Table 2 of the manuscript.**
(DOCX)

**S2 Table. Parameters and inputs comprising the 32 univariate sensitivity analyses, by study city.**
(DOCX)

**S3 Table. Benefit:Cost ratios and the *change* in the benefit term (when compared with the respective primary analysis) for the 32 univariate sensitivity analyses, by study city.** For comparison, the benefit:cost ratios in the primary analyses for Victoria, Kelowna and Halifax were 1.7:1, 1.9:1 and 2.1:1, respectively.
(DOCX)

## Acknowledgments

We appreciate the time of our city partners in verifying estimates of costs of active transportation investments in the study cities, as well as the broader IBIMS research team for their guidance and support on specific HEAT inputs.

## Author Contributions

**Conceptualization:** David G. T. Whitehurst, Meghan Winters.

**Formal analysis:** David G. T. Whitehurst, Danielle N. DeVries.

**Funding acquisition:** David G. T. Whitehurst, Daniel Fuller, Meghan Winters.

**Methodology:** David G. T. Whitehurst, Danielle N. DeVries, Daniel Fuller, Meghan Winters.

**Writing – original draft:** David G. T. Whitehurst, Danielle N. DeVries.

**Writing – review & editing:** David G. T. Whitehurst, Danielle N. DeVries, Daniel Fuller, Meghan Winters.

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
