## [Decision Letter · Decision Letter 0]

16 Sep 2020

PONE-D-20-15052

An economic analysis of the health-related benefits associated with bicycle infrastructure investment in three Canadian cities: an application of the World Health Organization’s Health Economic Assessment Tool

PLOS ONE

Dear Dr. David GT Whitehurst

Thank you for submitting your manuscript to PLOS ONE. After careful consideration, we feel that it has merit but does not fully meet PLOS ONE’s publication criteria as it currently stands. Therefore, we invite you to submit a revised version of the manuscript that addresses the points raised during the review process.

We look forward to receiving your revised manuscript.

Kind regards,

Carlos Alberto Zúniga-González, Ph.D

Academic Editor

PLOS ONE

Journal Requirements:

Additional Editor Comments (if provided):

Dear author I think that it a good work, however is important to spread details for the results comment and discussion. 

Reviewers' comments:

Reviewer's Responses to Questions

**Comments to the Author**

1. Is the manuscript technically sound, and do the data support the conclusions?

Reviewer #1: Yes

Reviewer #2: Yes

2. Has the statistical analysis been performed appropriately and rigorously? 

Reviewer #1: Yes

Reviewer #2: Yes

3. Have the authors made all data underlying the findings in their manuscript fully available?

Reviewer #1: Yes

Reviewer #2: Yes

4. Is the manuscript presented in an intelligible fashion and written in standard English?

Reviewer #1: Yes

Reviewer #2: Yes

5. Review Comments to the Author

Reviewer #1: In this paper, an economic analysis has been performed to identify the benefits of infrastructure investment in three Canadian cities. For this paper, World Health Organization’s health economic assessment tool has been used. The topic is interesting, and the paper is well-written. However, several simplified assumptions have been made in the study. For example, the future advancements in technology such as the autonomous vehicles or hybrid vehicles has been ignored. It would be suggested to add a section discussing the shortcomings and limitations of this research and suggestions for future research. Some other comments:

The full title seems to be long. It would be suggested to shorten it. You may also want to remove the second part.

One of the limitations in data adjustments is that it was assumed no temporal or spatial adjustments were needed. It should be noted that transportation-related data may always show temporal or spatial instability. For example, in term of safety, previous studies have shown that crash-related data show temporal instability across different time periods. In addition to safety, other elements studied in this research may show temporal/spatial instability. It would be suggested to add this point as one of the limitations of the study.

Reviewer #2: After reading the paper with high interest I think it covers an interesting topic and presents a simple, but straightforward, aim, methodology and results with high concerns for policy makers. However, from my view, the paper presents some gaps that might be revised before being published. Mainly, the literature review is scarce, and introduction and discussion should be broadened to cope with novelties and contributions of the paper regarding the existing literature.

Main comments:

• Introduction. The literature review is scarce, and it should be expanded, highlighting the novelties that the present paper shows and the contributions it is expected to address.

• Discussion. Some points are underlined in this section.

(1) The discussion section does not analyse/discuss the results in depth, neither the implications of the review for policy and management have been tackled. The authors focus on the implications of using HEAT too much, instead of deriving the recommendations for policy making. Actually, they expose just some general statements (lines 290, 344-345). Thus, the authors are encouraged to include the main policy suggestions and recommendations that are derived from the results, applied to the specific cities that they have assessed.

(2) As the authors state, one of the strengths of the work is the collaboration with stakeholders (line 350). However, along the paper there is no references to any stakeholder consultation. Please, in case it does, it should be included in the Methods section (what type of stakeholder has been consulted, how many, when…), and also the main statements derived from this consultation.

• Conclusion. This section is quite general. Please, rewrite it focusing on the main and concrete statements can be derived from the results and the revised discussion.

Minor comments:

• Please, rename the first section as “Introduction” instead of “Background”.

6. PLOS authors have the option to publish the peer review history of their article (what does this mean?). If published, this will include your full peer review and any attached files.

Reviewer #1: No

Reviewer #2: No

---

## [Author Response · Author response to Decision Letter 0]

23 Oct 2020

We thank the reviewers for their positive and constructive comments. We've included an attached file that details are our point-by-point responses, highlighting any changes that have been made to the manuscript. We hope that the amendments meet with the approval of the reviewers and the Editorial Board and look forward to your response.

---

## [Decision Letter · Decision Letter 1]

20 Jan 2021

An economic analysis of the health-related benefits associated with bicycle infrastructure investment in three Canadian cities

PONE-D-20-15052R1

Dear Dr. David GT Whitehurst

We’re pleased to inform you that your manuscript has been judged scientifically suitable for publication and will be formally accepted for publication once it meets all outstanding technical requirements.

Kind regards,

Carlos Alberto Zúniga-González, Ph.D

Academic Editor

PLOS ONE

Additional Editor Comments (optional):

Reviewers' comments:

Reviewer's Responses to Questions

**Comments to the Author**

1. If the authors have adequately addressed your comments raised in a previous round of review and you feel that this manuscript is now acceptable for publication, you may indicate that here to bypass the “Comments to the Author” section, enter your conflict of interest statement in the “Confidential to Editor” section, and submit your "Accept" recommendation.

Reviewer #1: All comments have been addressed

Reviewer #2: All comments have been addressed

2. Is the manuscript technically sound, and do the data support the conclusions?

Reviewer #1: Yes

Reviewer #2: Yes

3. Has the statistical analysis been performed appropriately and rigorously? 

Reviewer #1: Yes

Reviewer #2: Yes

4. Have the authors made all data underlying the findings in their manuscript fully available?

Reviewer #1: Yes

Reviewer #2: Yes

5. Is the manuscript presented in an intelligible fashion and written in standard English?

Reviewer #1: Yes

Reviewer #2: Yes

6. Review Comments to the Author

Reviewer #1: I would like to thank the authors for their efforts in addressing the comments raised in the first round of review.

Reviewer #2: (No Response)

7. PLOS authors have the option to publish the peer review history of their article (what does this mean?). If published, this will include your full peer review and any attached files.

Reviewer #1: No

Reviewer #2: No

---

## [Editor Report · Acceptance letter]

25 Jan 2021

PONE-D-20-15052R1 

An economic analysis of the health-related benefits associated with bicycle infrastructure investment in three Canadian cities 

Dear Dr. Whitehurst:

I'm pleased to inform you that your manuscript has been deemed suitable for publication in PLOS ONE. Congratulations! Your manuscript is now with our production department. 

Kind regards, 

on behalf of

Dr. Prof. Carlos Alberto Zúniga-González 

Academic Editor

PLOS ONE